# Optimizing SSVEP-Based BCI System towards Practical High-Speed Spelling

**DOI:** 10.3390/s20154186

**Published:** 2020-07-28

**Authors:** Jiabei Tang, Minpeng Xu, Jin Han, Miao Liu, Tingfei Dai, Shanguang Chen, Dong Ming

**Affiliations:** 1Lab of Neural Engineering & Rehabilitation, Department of Biomedical Engineering, School of Precision Instruments and Optoelectronics Engineering, Tianjin University, Tianjin 300072, China; jiahe@tju.edu.cn (J.T.); minpeng.xu@tju.edu.cn (M.X.); jinhan@tju.edu.cn (J.H.); daitingfei@tju.edu.cn (T.D.); shanguang_chen@126.com (S.C.); 2Tianjin International Joint Research Center for Neural Engineering, Academy of Medical Engineering and Translational Medicine, Tianjin University, Tianjin 300072, China; 3005202010@tju.edu.cn; 3National Key Laboratory of Human Factors Engineering, China Astronaut Research and Training Center, Beijing 100094, China

**Keywords:** brain–computer interface (BCI), steady-state visual evoked potential (SSVEP), practical, dynamic stopping (DS), modified task-related component analysis (mTRCA)

## Abstract

The brain–computer interface (BCI) spellers based on steady-state visual evoked potentials (SSVEPs) have recently been widely investigated for their high information transfer rates (ITRs). This paper aims to improve the practicability of the SSVEP-BCIs for high-speed spelling. The system acquired the electroencephalogram (EEG) data from a self-developed dedicated EEG device and the stimulation was arranged as a keyboard. The task-related component analysis (TRCA) spatial filter was modified (mTRCA) for target classification and showed significantly higher performance compared with the original TRCA in the offline analysis. In the online system, the dynamic stopping (DS) strategy based on Bayesian posterior probability was utilized to realize alterable stimulating time. In addition, the temporal filtering process and the programs were optimized to facilitate the online DS operation. Notably, the online ITR reached 330.4 ± 45.4 bits/min on average, which is significantly higher than that of fixed stopping (FS) strategy, and the peak value of 420.2 bits/min is the highest online spelling ITR with a SSVEP-BCI up to now. The proposed system with portable EEG acquisition, friendly interaction, and alterable time of command output provides more flexibility for SSVEP-based BCIs and is promising for practical high-speed spelling.

## 1. Introduction

Brain–computer interfaces (BCIs) allow users to communicate with the external devices by converting brain signals into commands [1,2]. BCIs can help people with neuromuscular diseases to improve the life quality [3] or help special appliance operators like astronauts whose movements were restricted by the environment to work more efficiently [4,5]. As a kind of brain signal that owns high temporal resolution and convenience of acquisition, electroencephalogram (EEG) is welcomed by BCI researchers. Event-related potentials (ERPs) [6,7], steady-state visual evoked potentials (SSVEPs) [8,9], and event-related desynchronization/synchronization (ERD/ERS) [10,11] are typical EEG features used in BCI researches.

Of these features SSVEPs that were induced by repetitive stimuli are widely employed in cognitive research and high-speed BCI systems for their high stability and signal-to-noise ratio (SNR) [12,13]. Researchers payed great efforts on the performance improvement of SSVEP-based BCIs in recent years, which mainly focused on the number of targets and target recognition algorithms. In order to increase the number of targets, a variety of novel coding methods were proposed, e.g., frequency shift keying (FSK) method that encodes commands into binary digits with two frequencies [14], intermodulation frequencies method that uses additional modulation frequencies [15], and hybrid coding methods that combine other EEG features such as P300 [16,17], etc. In particular, joint frequency-phase modulation (JFPM) method has been proved to improve the separability between targets and achieve high-speed SSVEP-BCI systems [18]. With regard to algorithms, various kinds of identification algorithms were applied in the SSVEPs-based BCIs [19], e.g., canonical correlation analysis (CCA) [20] and its various optimizations [21,22], multivariate synchronization index (MSI) [23], maximal-phase-locking value and minimal-distance (MP and MD, respectively) [24], and task-related component analysis (TRCA) [25], etc. Thanks to the endeavor in the encoding and decoding methods, the SSVEP-based BCIs have achieved the highest information transfer rate (ITR) among the noninvasive BCI paradigms.

Although many high-speed systems based on SSVEPs have been established in previous studies, it is meaningful to enhance the practicability of the systems for spelling in real life. For example, most of these studies used EEG devices designed for research in the laboratory [9,21,22,25], such as Neuroscan Synamps2 system. The research-grade devices possess excellent signal amplification performance and diverse functions, whereas most of the functions are superfluous for a practical BCI system and push up the cost. In addition, the 5 × 8 matrix layout was popular in previous SSVEP-based spellers and the users needed to remember the location of each command before the experiment, which increases the workload and slows down the spelling. Furthermore, these systems used fixed stimulating time, i.e., the fixed stopping (FS) strategy. As is known in the P300-based BCIs, the dynamic stopping (DS) strategy enables the self-check of recognition confidence for a BCI system so as to quicken the output when it is confident about the result, whereas keep acquiring data when the correctness of the decision is not sure [26,27]. A few studies have tested the performance of DS strategy in an SSVEP-based BCI [28,29,30], but it was evaluated by offline analyzing the offline collected EEG data like an online experiment. The feasibility of real-time DS strategy in a practical online system needs further verification.

The goal of this study was to design a high-speed SSVEP-based BCI system for practical use. Figure 1 is the block diagram of our system. The novelty of our system was reflected from several aspects. Firstly, we simplified the acquisition by developing a dedicated low-cost EEG amplifier with self-designed circuit and optimized the stimulation by arranging the instructions like a keyboard that would be familiar to users. Secondly, in order to extract the SSVEPs more effectively for a high-speed system, the standard forward filtering was applied instead of the frequently-used zero-phase filtering for reliable online noise reduction and the TRCA spatial filter was modified (named mTRCA) to enhance the target recognition. Last, but not the least, the DS strategy based on Bayesian posterior probability was incorporated into the system to obtain flexible stimulating time and improve the ITRs.

As the trial duration of a SSVEP-BCI is much shorter than that of a P300-BCI due to the different coding schemes, three issues were concerned for real-time DS in an online SSVEP-BCI. The first concern is the unfixed stimulating time caused by the immediate stopping of stimulus after satisfying the output condition in DS strategy. As most users have been used to fixed stimulating time in normal FS BCIs, the sudden stopping of stimulating might distract the attention of users and delay the shifting to the next target; thus, leading to the performance decline. Therefore, this study used unfixed stimulating time in the offline calibration experiment to imitate the DS in online operation so that the subjects could adjust to the unfixed timing, and the variation of EEGs could also be covered by the calibration data. The second concern focused on the output condition of DS due to the fact that the probability distribution might change with the stimulus frequency and data length. We raised an adaptive threshold generating method that was easy to implement so as to fit the variation of probability distribution. Another concern is that the real-time DS requires the system to perform the recognition algorithm in very short time. To this end, the programs of some key processes for recognition were ported to C Mex from MATLAB for accelerating execution to ensure the real-time performance.

## 2. Materials and Methods

### 2.1. Experimental Protocol

#### 2.1.1. Stimulus Design for Practical Spelling

Figure 2a illustrates the design of the stimulation. The participants were seated at a distance of 60 cm from a liquid-crystal display (LCD) monitor with the refresh rate of 60 Hz. In order to make the system friendly and practical for users, forty targets were rearranged as the pattern of a keyboard with each target subtended 2° of visual angle. In particular, the keys of Backspace and Space were set longer than on general keyboards. An output box was placed above the targets. The frequencies ranged from 8.0 to 15.8 Hz with an interval of 0.2 Hz and the phase interval between two neighboring frequencies was 0.35 π, which were in accordance with the JFPM method in previous studies. In addition, the frequency approximation approach proposed by Wang et al. [31] was used to modulate frequencies and phases in the monitor. The stimulation was developed on the MATLAB platform using the Psychtoolbox 3 [32], and the stimulating onset triggers were sent to the EEG amplifier via user datagram protocol (UDP).

#### 2.1.2. Experimental Procedure

Twelve healthy subjects (five males and seven females) aged 20 to 26 years old with normal or corrected normal sight participated in this study. The study was conducted in accordance with the Declaration of Helsinki and the experimental procedures were approved by the Institutional Review Board at Tianjin University. The participants provided written consent after the details of the experiment were explained. All the subjects participated in both the offline and online experiments.

Figure 2b shows the trial timing of the experiments. Each trial started with a rest period for 0.5 s, followed by a flash stage. A yellow box would appear around the target as a cue. The subjects were asked to shift their gaze to the target as soon as possible within the rest stage and focus on the dot displayed at the center of the target within the flash stage. In order to acquire a model that were fit with the unfixed stimulating time in DS situation, three kinds of stimulating time (0.32 s, 0.6 s, and 1 s) were randomly used in offline stimulation. The offline calibration experiment consisted of six rounds, with three kinds of stimulating time used for each of the 40 targets per round. Hence, the total trials were 40 × 3 × 6 = 720 and the total experimental time was 0.5 × 720 + (0.32 + 0.6 + 1) × 40 × 6 = 13.68 min. The 720 trials were divided into 12 blocks with 60 trials in each block. After offline blocks, we could obtain 18 EEG epochs between 0.2 s to 0.32 s, 12 epochs between 0.32 s to 0.6 s, and 6 epochs between 0.6 s to 1 s for each target, as shown in Figure 2c.

The online experiment contained two sub-experiments. Firstly, a cue-guided experiment including 5 blocks was conducted and the subjects were asked to complete 40 trials corresponding to all 40 targets in each block. When a stopping trigger was received from the algorithm (introduced in 2.4) during the flash stage, the stimulation program would stop flashing and prompt the next target with the result displayed in the output box at the same time. If the stimulating time reached 1 s without a stopping trigger (trial 3 in Figure 2b), the program would treat the target as incorrect and prompt the next one. A free spelling experiment was conducted following the cue-guided experiment. The subjects were asked to input “TIANJIN UNIVERSITY 1895” two times without visual cues. The result of each trial would be displayed in the output box and reported by voice as feedback. When an incorrect input happened, the subjects should stare at the key of Backspace to remove the wrong character. The gaze shifting time was determined according to the self-feeling of subjects after a tentative block was completed. 

### 2.2. EEG Recording and Preprocessing

#### 2.2.1. EEG Acquisition System

The lower right of Figure 1 shows the circuit of the amplifier developed in this study. The device was designed based on the analog front-end ADS1299 (Texas Instruments, Dallas, TX, USA), which owns fine resolution delta-sigma ADC (24-bit), high common-mode rejection ratio (CMRR = −110 dB), and low input-referred noise (1 μV). The chip has the function of electrode impedance measurement that is important for EEG acquisition. Each ADS1299 supports up to 8 channels and two chips were used in a daisy-chain configuration for 16-channel data collection in our research. The ADS1299 was controlled by a STM32F407VET6 processor through serial peripheral interface (SPI). A W5500 ethernet module was utilized to connect the processor with the computer via the local area network (LAN). The system was powered by a Li-polymer Battery (Zhenfa ZF-103450, 2000 mAh). A metal shell was used to package the device.

We designed a 16-channel EEG cap with Ag/AgCl electrodes placed at standard positions of international 10–20 system. All channels were referenced to the vertex and grounded to prefrontal lobe between FPz and Fz during acquisition. The EEG data from eleven channels around the occipital area (P5, Pz, P6, PO5, PO3, POz, PO4, PO6, O1, Oz, and O2) were used for analyses and online tests. The EEG signals were sampled at 250 Hz and transmitted to the computer through transmission control protocol/internet protocol (TCP/IP) for storing and analyzing. 

#### 2.2.2. EEG Preprocessing

In pre-processing, the data were notch filtered at 50 Hz and band-pass filtered between (m×9−2) Hz and 90 Hz according to the filter bank strategy (*m* = 1,2,...,8 in this study). The filter parameters were generated with Chebyshev Type I infinite impulse response (IIR) filter design method. The EEG epochs were extracted in [0.14 s, 0.14 + *t* s] according to the onset triggers sent by the stimulation program, with the latency delay in the visual system defined as 0.14 s.

### 2.3. Target Recognition Algorithm

#### 2.3.1. Modified TRCA-Based Spatial Filter

For an EEG epoch X=(x1,x2,⋯,xNc)T∈ℝNc×Nt, the spatial filtering process is to get a linear sum of all channels:(1)y=wTX=∑k=1NcwkxkT∈ℝ1×Nt

Here, $N_{c}$ indicates the number of channels, $N_{t}$ is the number of sampling points and $w={\left(w_{1},w_{2},\cdots ,w_{N_{c}}\right)}^{T}$ is the spatial filter vector. The spatial filter generated by TRCA has shown excellent performance in recent SSVEP-based BCI systems [25,30,33]. For frequency *i*, the TRCA aims to maximize the reproducibility from trial to trial:(2)$$\begin{array}{ll} C_{i} & =\frac{1}{N_{i}\left(N_{i}-1\right)}\sum_{h_{1}=1}^{N_{i}}\sum_{\begin{array}{l} h_{2}=1\\ h_{1}\neq h_{2} \end{array}}^{N_{i}}\mathrm{cov}\left(y_{i}^{\left(h_{1}\right)},y_{i}^{\left(h_{2}\right)}\right)=\frac{1}{N_{i}\left(N_{i}-1\right)}\sum_{h_{1}=1}^{N_{i}}\sum_{\begin{array}{l} h_{2}=1\\ h_{1}\neq h_{2} \end{array}}^{N_{i}}y_{i}^{\left(h_{1}\right)}y_{i}^{\left(h_{2}\right)}{}^{T}\\ & =\frac{1}{N_{i}\left(N_{i}-1\right)}\sum_{h_{1}=1}^{N_{i}}\sum_{\begin{array}{l} h_{2}=1\\ h_{1}\neq h_{2} \end{array}}^{N_{i}}\left[w_{i}{}^{T}X_{i}^{\left(h_{1}\right)}\right]{\left[w_{i}{}^{T}X_{i}^{\left(h_{2}\right)}\right]}^{T}=\frac{1}{N_{i}\left(N_{i}-1\right)}\sum_{h_{1}=1}^{N_{i}}\sum_{\begin{array}{l} h_{2}=1\\ h_{1}\neq h_{2} \end{array}}^{N_{i}}w_{i}{}^{T}X_{i}^{\left(h_{1}\right)}X_{i}^{\left(h_{2}\right)}{}^{T}w_{i}\\ & =w_{i}{}^{T}\left[\frac{1}{N_{i}\left(N_{i}-1\right)}\sum_{h_{1}=1}^{N_{i}}\sum_{\begin{array}{l} h_{2}=1\\ h_{1}\neq h_{2} \end{array}}^{N_{i}}X_{i}^{\left(h_{1}\right)}X_{i}^{\left(h_{2}\right)}{}^{T}\right]w_{i}=w_{i}{}^{T}S_{i}w_{i}\to \max \end{array}$$
where *h* indicates the index of training trials, and $N_{i}$ is the number of training trials. The *S_i_* could be written as follows:(3)$$\begin{array}{ll} S_{i} & =\frac{1}{N_{i}\left(N_{i}-1\right)}\sum_{h_{1}=1}^{N_{i}}\sum_{\begin{array}{l} h_{2}=1\\ h_{1}\neq h_{2} \end{array}}^{N_{i}}X_{i}^{\left(h_{1}\right)}X_{i}^{\left(h_{2}\right)}{}^{T}=\frac{1}{N_{i}\left(N_{i}-1\right)}\left[\sum_{h_{1}=1}^{N_{i}}\sum_{h_{2}=1}^{N_{i}}X_{i}^{\left(h_{1}\right)}X_{i}^{\left(h_{2}\right)}{}^{T}-\sum_{h_{1}=1}^{N_{i}}\sum_{\begin{array}{l} h_{2}=1\\ h_{1}=h_{2} \end{array}}^{N_{i}}X_{i}^{\left(h_{1}\right)}X_{i}^{\left(h_{2}\right)}{}^{T}\right]\\ & =\frac{1}{N_{i}-1}\left[\frac{1}{N_{i}}\sum_{h_{1}=1}^{N_{i}}X_{i}^{\left(h_{1}\right)}\sum_{h_{2}=1}^{N_{i}}X_{i}^{\left(h_{2}\right)}{}^{T}-\frac{1}{N_{i}}\sum_{h=1}^{N_{i}}X_{i}^{\left(h\right)}X_{i}^{\left(h\right)}{}^{T}\right]=\frac{1}{N_{i}-1}\left[N_{i}\cdot {\bar{X}}_{i}{\bar{X}}_{i}{}^{T}-Q_{i}\right] \end{array}$$
in which ${\bar{X}}_{i}=\frac{1}{N_{i}}\sum_{h=1}^{N_{i}}X_{i}^{\left(h\right)}$ represents the average across trials, and $Q_{i}=\frac{1}{N_{i}}\sum_{h=1}^{N_{i}}X_{i}^{\left(h\right)}X_{i}^{\left(h\right)}{}^{T}$ is then used to constrain the variance that satisfies $\mathrm{Var}\left(y\right)=w_{i}{}^{T}Q_{i}w_{i}=1$.The TRCA can be formulated as an eigenvalue problem,
(4)$${\hat{w}}_{i}=\underset{w_{i}}{\arg\max}\frac{w_{i}{}^{T}S_{i}w_{i}}{w_{i}{}^{T}Q_{i}w_{i}}$$

The aim of TRCA is to the maximize of intra-class correlation of each frequency. If the spatial filter could also minimize the inter-class correlation between the current frequency and other frequencies, the risk of misclassifying this frequency to the another one would be reduced. Hence, a modification was conducted by employing the covariance $C_{ij}$ between frequency i and j(j≠i) in this study,
(5)$$\begin{array}{ll} C_{ij} & =\frac{1}{N_{i}N_{j}}\sum_{h_{1}=1}^{N_{i}}\sum_{h_{2}=1}^{N_{j}}\mathrm{cov}\left(y_{i}^{\left(h_{1}\right)},y_{j}^{\left(h_{2}\right)}\right)=\frac{1}{N_{i}N_{j}}\sum_{h_{1}=1}^{N_{i}}\sum_{h_{2}=1}^{N_{j}}w_{i}{}^{T}\left[X_{i}^{\left(h_{1}\right)}X_{j}^{\left(h_{2}\right)}{}^{T}+X_{j}^{\left(h_{2}\right)}X_{i}^{\left(h_{1}\right)}{}^{T}\right]w_{i}\\ & =w_{i}{}^{T}\left[\sum_{h_{1}=1}^{N_{i}}\sum_{h_{2}=1}^{N_{j}}\frac{1}{N_{i}N_{j}}\left(X_{i}^{\left(h_{1}\right)}X_{j}^{\left(h_{2}\right)}{}^{T}+X_{j}^{\left(h_{2}\right)}X_{i}^{\left(h_{1}\right)}{}^{T}\right)\right]w_{i}=w_{i}{}^{T}S_{ij}w_{i}\to \min \end{array}$$
where $S_{ij}$ is defined as
(6)$$S_{ij}=\sum_{h_{1}=1}^{N_{i}}\sum_{h_{2}=1}^{N_{j}}\frac{1}{N_{i}N_{j}}\left(X_{i}^{\left(h_{1}\right)}X_{j}^{\left(h_{2}\right)}{}^{T}+X_{j}^{\left(h_{2}\right)}X_{i}^{\left(h_{1}\right)}{}^{T}\right)={\bar{X}}_{i}{\bar{X}}_{j}{}^{T}+{\bar{X}}_{j}{\bar{X}}_{i}{}^{T}$$

Then the covariance matrix $S_{i}$ is modified as
(7)$$S_{i}^{\prime }=2S_{i}-\frac{1}{N_{f}-1}\sum_{j=1,j\neq i}^{N_{f}}S_{ij}$$
and the matrix $Q_{i}$ could be modified accordingly as
(8)$$Q_{i}^{\prime }=\frac{1}{2}\left(Q_{i}+\frac{1}{N_{f}-1}\sum_{j=1,j\neq i}^{N_{f}}Q_{j}\right)$$

In this way, the mTRCA spatial filter ${\hat{w}}_{i}$ could be derived from Equation (9) as the eigenvectors of ${Q_{i}^{\prime }}^{-1}S_{i}^{\prime }$ by solving the eigenvalue decomposition problem,
(9)$${\hat{w}}_{i}=\underset{w_{i}}{\arg\max}\frac{w_{i}{}^{T}S_{i}^{\prime }w_{i}}{w_{i}{}^{T}Q_{i}^{\prime }w_{i}}$$

#### 2.3.2. The mTRCA-Based Decoder

The spatial filter wi(m) (i=1,2,…,Nf) were constructed for *m*-th sub-band based on the above methods, followed by the ensemble of all spatial filters [25] as
(10)W(m)=[w1(m),w2(m),…,wNf(m)]∈ℝNc×Nf

Then the average training data across trials of the *i*-th frequency ${\bar{\chi }}_{i}^{\left(m\right)}$ would be multiplied by $W^{\left(m\right)}$ as the template. When a testing EEG epoch $X^{\left(Test\right)}$ was acquired, it would be temporally filtered and spatially filtered, followed by the Pearson correlation coefficients with the templates calculated as $r_{i}^{\left(m\right)}$. The final coefficients $r_{i}$ were calculated by a weighted mean of the coefficients corresponding to all sub-bands, as shown in Figure 3.

### 2.4. DS Strategy

#### 2.4.1. Probabilistic Model for DS Strategy

The Bayesian-based methods are commonly used in previous DS studies [29,30,34]. This study followed this idea and proposed optimized procedures to construct a probabilistic model. For the training data of frequency *j*, the correlation coefficients $r_{ji}$ could be calculated from Decoder [*i*,*t*] when the data length was *t*. The coefficients were then normalized with the z-score method,
(11)$${\tilde{r}}_{ij}=\frac{r_{ij}-\mathrm{mean}(r_{ij})}{\mathrm{std}(r_{ij})}$$
where
(12)$$\mathrm{mean}(r_{ij})=\frac{1}{N_{f}}\sum_{i=1}^{N_{f}}r_{ij},\;\mathrm{std}(r_{ij})=\sqrt{\frac{\sum_{i=1}^{N_{f}}{\left[r_{ij}-\mathrm{mean}(r_{ij})\right]}^{2}}{N_{f}-1}}$$
represents the average and standard deviation of the coefficients. In this way, we could obtain $N_{f}\times N_{f}$ kinds of coefficients as shown in Figure 4a. Suppose the correct prediction is written as $H_{1}$, while the incorrect prediction is written as $H_{0}$. For Decoder [*i*,*t*], the likelihood probability density functions (pdfs) of $H_{0}$ and $H_{1}$ could be estimated through Gaussian kernel density method as $P_{i}\left(r|H_{0},t\right)$ and $P_{i}\left(r|H_{1},t\right)$, respectively. 

In DS strategy, the command will be output if the probability reaches the threshold. However, the proper threshold for each target might change with the stimulus frequency, leading to unfitness of a fixed threshold in different cases. The grid-search method used in previous work [30] might be time consuming. This study proposed an adaptive threshold generating method by utilizing the pdfs of each decoder (Figure 4b). The correlation coefficients corresponding to the maximum of $P_{i}\left(r|H_{0},t\right)$ and $P_{i}\left(r|H_{1},t\right)$ were named as $r_{\max}^{H_{0}}$ and $r_{\max}^{H_{1}}$, respectively. The left and right boundary value were defined as the coefficients corresponding to a quarter of the maximum, and termed as $r_{L}^{H_{0}}$, $r_{R}^{H_{0}}$, $r_{L}^{H_{1}}$, and $r_{R}^{H_{1}}$, respectively. Then, the threshold of frequency i under data length of was defined as
(13)$$p_{i,t}^{\left(\mathrm{th}\right)}=1-\frac{d_{\max}}{d_{1}+d_{2}}=1-\frac{r_{\max}^{H_{1}}-r_{\max}^{H_{0}}}{\left(r_{R}^{H_{1}}-r_{L}^{H_{0}})+(r_{L}^{H_{0}}-r_{R}^{H_{1}}\right)}\in \left(0,1\right]$$

The dashed line in Figure 4b marks the value of threshold. When the separability between the two pdfs was low, as the upper right of Figure 4b shows, the threshold was set higher for the sake of conservation. Conversely, if the two pdfs lived far from each other, it is easier to make the right decision; hence, the threshold was set smaller as shown at the lower right of Figure 4b.

#### 2.4.2. Procedure of Online Target Recognition with DS

Figure 5 is the process diagram of online recognition with DS strategy. The data length was set as $t=t_{0}$ (200 ms in this study) at the beginning. The new EEG data were fed into the filter bank and Decoder [*i*,*t*] in turn to generate the correlation coefficients. After normalization, the ${\tilde{r}}_{i}$ were fed into the corresponding pdfs and we could get the probabilities $p\left({\tilde{r}}_{i}|H_{1},t\right)$ and $p\left({\tilde{r}}_{i}|H_{0},t\right)$. The posterior probabilities could be calculated via the Bayesian formula.
(14)$$p_{i}\left(H_{0}|\tilde{r},t\right)=\frac{p(H_{1})p\left({\tilde{r}}_{i}|H_{1},t\right)}{p(H_{1})p\left({\tilde{r}}_{i}|H_{1},t\right)+p(H_{0})p\left({\tilde{r}}_{i}|H_{0},t\right)}$$

The prior probabilities $p(H_{1})$ and $p(H_{0})$ were set as 0.5 in this study. If the maximum of the 40 posterior probabilities reached the threshold generated in Section 2.4, the character corresponding to the index of the maximum would be output and start the next spelling. Otherwise, the data length *t* would become *t* + Δ*t* and then repeated the procedure above when the length of new data reached *t* + Δ*t*. The step of data length Δ*t* was set as 20 ms. If no posterior probability reached the threshold after *t* = 1 s, no character would be output and the subject would begin the next spelling.

### 2.5. System Optimization for High-Speed Online Operation

#### 2.5.1. Program Optimization for Real-Time DS

As shown in Section 2.4, the online DS requires the recognition algorithm to be executed over and over again to compare the probabilities and the thresholds. Considering the code execution efficiency of C++ is higher than that of MATLAB, three key procedures of online recognition were reprogrammed with C++ and compiled into MATLAB executable (.mexw64) files [35] with C Matrix Library API and C MEX Library API of MATLAB in order to ensure the real-time performance:(Filter): including the 50 Hz notch filtering and the 8 bandpass filtering processes.(Corr.): including the spatially filtering and the calculation of 40 (frequencies) × 8 (bands) = 320 coefficients.(Prob.): including the normalization of correlation coefficients and the calculation of 40 posterior probabilities.

After the experiment, the elapsed time of these procedures with MATLAB and C Mex was evaluated through simulated online DS process of the online data. The simulated online program was entirely consistent with the real online program. The simulation was run on a universal laptop (Model: Lenovo Xiaoxin Chao 7000-13; CPU: Intel Core i7-8550U, 2.90 GHz; RAM: 16 GB).

#### 2.5.2. Filtering Strategy Optimization and Comparison

In most of the SSVEP-based BCI researches, the forward and reverse filtering is commonly used to achieve zero-phase filtering, which can be implemented using the filtfilt() function in MATLAB [8,25,30]. Whether this filtering process could satisfy the DS recognition of SSVEPs in a real-time BCI system has not been fully investigated. The standard filtering forward process according to the rational transfer function is the most direct method to realize the real-time filtering [36]. This study took the standard method as the filtering strategy in online experiments and compared three kinds of filtering strategies (Figure 6) through simulated online analysis of the online data after the experiment:(Filter): the standard forward filtering method using rational transfer function that could rolling update the filtered signal when a new sample point comes. The data length needed for filtering process prior to time *t* was equal to the order of filter *N*.(Filtfilt(−)): the forward and reverse filtering method combined with initial condition and signal extending, which was the same as the filtfilt() function. The data of *T*_(−)_ length prior to time *t* were used for filtering.(Filtfilt(−)(+)): the data of *T*_(−)_ length prior to time *t* and the data of *T*_(+)_ length posterior to time *t* were used for forward and reverse filtering.

Here, *T*_(−)_ = 3 s, *T*_(+)_ = 0.2 s. The three strategies were also reprogrammed with C Mex.

## 3. Results

### 3.1. Comparison of Performance with TRCA and mTRCA

Figure 7a,b display the averaged offline recognition accuracies and putative ITRs across all subjects with different data lengths using TRCA and mTRCA spatial filters. The accuracy and the ITR were estimated by a leave-one-out cross-validation. We used the Wilcoxon signed-rank test instead of paired *t*-test to compare the performance of the two methods. As a non-parametric statistical hypothesis test, this method could be an alternative to the paired *t*-test when the difference between two samples’ means does not satisfy normal distribution [37]. The mTRCA-based method achieved significantly higher accuracies than that of TRCA-based method, especially for data length ≤ 0.5 s. Note that the accuracies presented a small dip after data length exceeded 0.6 s, as only there were only six samples possessing data length of 0.6–1 s for each target (see Section 2.2 for details), which might pose an impact on the classification performance. The significance was consistent to that of the ITR. The highest ITR for mTRCA-based method was 293.5 ± 27.2 bits/min, which was significantly higher than 288.7 ± 26.8 bits/min for TRCA-based method (*t* = 0.4 s, *W*(12) = 75, *p* = 0.0024). We also verified the algorithm on the benchmark dataset proposed by Want et al. [38], as shown in Figure 7c,d. The mTRCA also outperformed TRCA on the accuracies and ITRs, though the improvement was not as significant as that of our experimental data. The possible reason for this difference is that each target had only six trials in the benchmark dataset, which was fewer than those in our offline data and led to insufficient training.

### 3.2. Online Performance with DS Strategy

Table 1 lists the results of online cued-guided experiments with the mTRCA-based algorithm. As the offline data were analyzed with fixed data length, the maximal offline ITRs were listed as the performance of FS strategy for comparison. Wilcoxon signed-rank tests indicated that the DS strategy significantly reduced the data length used for recognition (*W*(12) = 0, *p* = 1.2 × 10^−4^). Although the accuracies showed slightly decline without significance (*W*(12) = 31, *p* = 0.1937), the DS strategy significantly improved the ITRs (*W*(12) = 78, *p* = 4.9 × 10^−4^). The minimal and maximal ITRs were 260.6 bits/min and 420.2 bits/min, respectively.

Table 2 lists the results of online free spelling experiments. The spelling rate reached 38.2 ± 4.0 characters per minute (cpm) on average with a peak of 47.1 cpm. Although the ITRs were lower than those of cued-guided experiment due to the prolonged gaze shifting time, it is much closer to the practical situation, as the users have difficulty to shift their gaze in 0.5 s while considering the next character. The results demonstrated the effectiveness of the real-time DS strategy in an online SSVEP-based speller.

### 3.3. Comparison of Run Time between MATLAB and C Mex

The system executed hundreds of times of recognitions for the data of each subject. The elapsed time records were firstly averaged for each subject and then averaged across all subjects. Figure 8 shows the averaged run time of the three key procedures after a simulation of online recognition with MATLAB and C Mex, respectively. All the key procedures consumed significantly less time using C Mex than those of MATLAB (paired *t*-test, *p* < 0.001). Notably, the summation of the three procedures was 4.76 ± 0.41 ms with C Mex, which was significantly shorter than the Δ*t* of 20 ms in this study (*t*(11) = −127.67, *p* = 8.53 × 10^−9^), whereas it was 19.72 ± 0.87 ms with MATLAB, which was close to the Δ*t* (*t*(11) = −1.09, *p* = 0.298).

### 3.4. Comparison of Online Filtering Strategies

Figure 9 displays the performance with three filtering strategies. The Filtfilt(−) strategy showed significant lower accuracy than that of the Filter strategy (81.6% ± 9.1% vs. 87.5% ± 5.3%, *W*(12) = 2, *p* = 0.0015), which resulted in the decrease of ITR (295.3 ± 57.8 bits/min vs 330.4 ± 45.4 bits/min, *W*(12) = 1, *p* = 9.7 × 10^−4^). The accuracy of the Filtfilt(−)(+) strategy was similar to that of the Filter strategy (87.7% ± 6.9%, *W*(12) = 45, *p* = 0.3184), yet the ITR was also significantly lower (265.1 ± 41.2 bits/min, *W*(12) = 0, *p* = 4.9 × 10^−4^) due to the increased data length (444.9 ± 21.8 ms vs. 254.2 ± 27.7 ms, *W*(12) = 0, *p* = 4.9 × 10^−4^). Moreover, the elapsed time of both the Filtfilt(−) and Filtfilt(−)(+) strategies were longer than that of the Filter strategy (*W*(12) = 0, *p* = 4.9 × 10^−4^).

## 4. Discussion

The high-speed BCI systems based on SSVEPs attracted growing attention and stronger demand of applying this paradigm in daily life in recent years. This study optimized a high-speed SSVEP-based speller towards practical application. The speller was designed according to the layout of a keyboard so that it is convenient for users to find the intended character. The EEG data was acquired using a dedicated amplifier developed in our laboratory instead of the research-grade system in previous high-speed BCI studies. In order to provide flexible stimulating time, we incorporated the DS strategy based on Bayesian posterior probability into the online SSVEP-BCI. The introducing of above measures brought new problems to the BCI system, which need optimization from several perspectives.

The filtering process is our first concern for a high-speed BCI system. Previous studies have barely discussed the details of online filtering in a SSVEP-based system. As is known, the standard digital filtering is a convolution of the input signal with the impulse response of a digital filter. The IIR filters are the digital form of analog filters and welcomed in real-time applications for their low time delay. However, they cannot realize exact linear phase like finite impulse response (FIR) filters; thus, distorting the EEG signal and degrading the BCI performance. The forward and reverse filtering is a commonly used method to achieve the zero-phase filtering [39]. However, it is a noncausal filtering process that we need the signal prior and posterior to the current time for filtering and take out the useful signal afterwards. This will cause the problem that no signal exists posterior to the current time in a real-time system, leading to the distortion of the signals near the current time owing to the transient response at the beginning of filtering. In the filtfilt() function of MATLAB, the initial condition of the filter and the signal extending method were employed to mitigate the distortion [40,41]. Nevertheless, the comparison in Section 3.4 indicates that these methods could not compensate the loss of accuracy caused by the transient response (the Filtfilt(−) in Figure 9). If we use the posterior signal as the Filtfilt(−)(+) in Figure 9, the accuracy would be higher whereas the longer data length declined the ITR. Hence, the classical forward filtering method (Filter in Figure 9) is more suitable for the high-speed operation.

Another important part of work focused on the implementation of DS strategy. Although it has been applied in the P300-based BCI systems and significantly reduced the number of stimulating rounds for the output [34,42,43], the implementation of DS faces new challenges in an online SSVEP-based system. First, as illustrated in the introduction, the unfixed stimulating time in the DS situation results in different feelings for users compared with those in FS situation. Therefore, the offline experiment of this study was designed using three kinds of alternant trial length to emulate the unfixed stimulating time in DS situation. This strategy also increased the training samples corresponding to short data length so that guaranteed the accuracies. Second, the online DS requires the system to process the data as quickly as possible. In other words, the system needs higher “temporal resolution” to provide the real-time performance. The resolution Δ*t* in this study was set at 20 ms, which meant the data transmission and recognition process should be completed in 20 ms. The self-developed amplifier could send the data packet every 4 ms, i.e., the sample point could be obtained by the online program immediately after being collected, while the Neuroscan system sends the data packet every 40 ms according to our previous testing. Hence, the dedicated amplifier provided flexible temporal resolution to enhance the real-time performance. As for the data processing, it is of great importance to assess the execution time of the program before experiment. It is known that the MATLAB used in many BCI applications is a kind of interpreted language. The code execution efficiency is limited to some degree compared with the compiled language, especially for the looping structure. The recognition of SSVEP-based BCIs contains lots of loops, e.g., the calculation of correlation coefficients contains 40 (frequencies) × 8 (bands) = 320 loops in this study. The three key procedures of recognition consumed nearly 20 ms in total using the MATLAB program (Figure 8). If considering the other procedures, the run time would exceed Δ*t* when executed in MATLAB, and the accumulation of the delay might collapse the system. This study reprogrammed the key procedures with C++ and compiled them into MATLAB executable files. The run time was reduced to 4.76 ms on average, which was far less than the MATLAB program. In future works, the calculation could be further accelerated with multiple threads or dedicated processors such as field programmable gate arrays (FPGAs); thus, improving the temporal resolution for higher real-time performance.

Considering that the lab-assembled EEG device and the non-zero-phase filtering might put an impact on the classification accuracy, we modified the TRCA spatial filter by subtracting covariance of other frequencies from the current frequency, which could reduce the interference of irrelevant information. This modification obtained significantly higher ITRs than those of TRCA and S8 achieved a highest ITR of 420.2 bits/min, which is the highest online ITR for SSVEP-based BCI to our knowledge.

Despite the various measures towards a practical SSVEP-BCI conducted in this study, it should be noted that more aspects need to be considered in future work. One of the most important issues is about analyzing the experience of the users from the perspective of psychology. For example, could a patient with physical impairments accept such a system [44,45]? Could the users adapt to this system as it works at a very high speed? If the system output several incorrect results, could the users avoid negative emotion and continue focusing on the intended command [46]? If not, there might be more and more wrong outputs and the human–computer system would collapse. As the AI has caused some ethical problems [47,48], the BCI might face similar situation someday. Such problems have not got enough attention to our knowledge. In future developments, it might be better to design some psychological programs for users such as a questionnaire or an interview about their feeling about the system, and to personalized optimize the system configuration so as to make the users work with the BCIs comfortably and efficiently.

## 5. Conclusions

This paper presents a series of improvement approaches for a more practical high-speed spelling system based on SSVEPs. The stimulation was designed as a keyboard instead of the commonly used matrix layout, and the EEG data were acquired from a self-developed EEG device instead of research-grade devices. Particularly, this study incorporated the Bayesian-based DS strategy into the online system; thus, realizing alterable stimulating time and higher ITRs. Considering the new challenges brought by above measures, the system was optimized from the aspects of temporal filter, spatial filter, calibration experiment, and programming. The proposed system achieved the highest online ITR reported in BCI speller studies to date, which demonstrated the feasibility of the proposed algorithm and system framework. This work provides methodological guidelines for designing high-speed SSVEP-based systems towards spelling in real life and is promising to develop more interesting and practical applications.

## Figures and Tables

**Figure 1 sensors-20-04186-f001:**
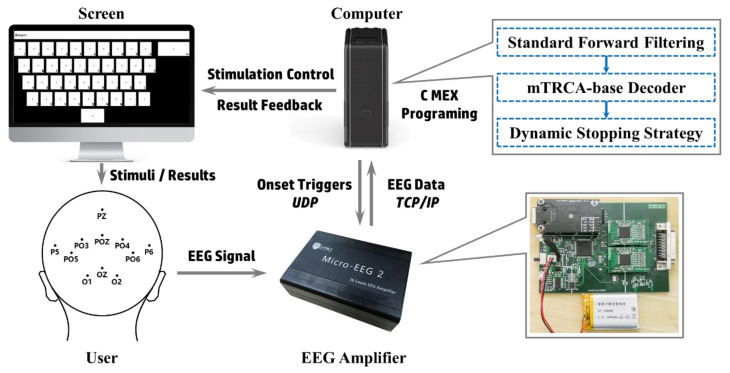
Overview of the system framework. The stimulation was controlled by a computer, and the electroencephalogram (EEGs) were acquired by a self-developed EEG amplifier. The stimulus onset triggers and the EEG data were sent and received with user datagram protocol (UDP) and transmission control protocol/internet protocol (TCP/IP), respectively. The EEG data were decoded by the computer and the results were presented on the screen. The lower right is the circuit board of the EEG amplifier, and the upper right shows the innovation of the EEG decoding.

**Figure 2 sensors-20-04186-f002:**
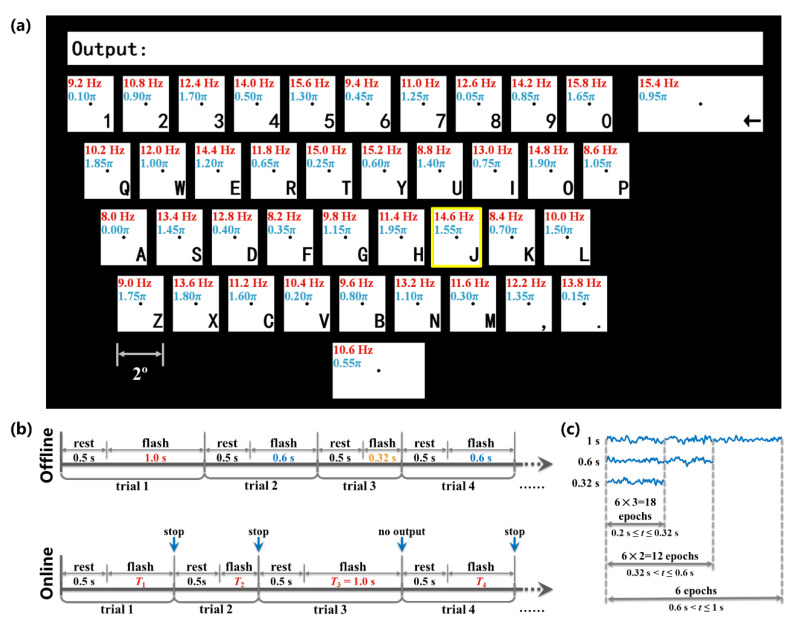
The stimulus design of the 40-target steady-state visual evoked potential (SSVEP)-based brain–computer interface (BCI) system. (**a**) Schematic of the stimulation, in which the 40 targets were distributed according to the layout of a keyboard and the frequency and phase of each target were marked out at upper left corner. The cue was presented to subjects as a yellow rectangle around the target. (**b**) Trial timing diagram of the experiment. (**c**) Three kinds of EEG epochs corresponding to different data lengths were obtained after offline experiment.

**Figure 3 sensors-20-04186-f003:**
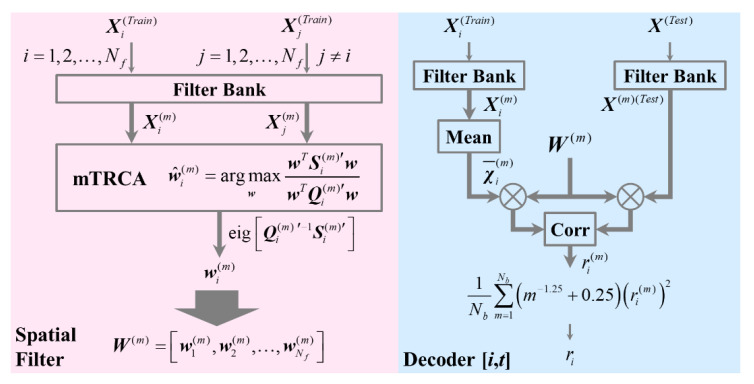
The process of constructing a model for target recognition: (**left**) the spatial filter based on modified task-related component analysis (mTRCA), (**right**) the decoder corresponding to the *i*-th frequency with data length of *t*.

**Figure 4 sensors-20-04186-f004:**
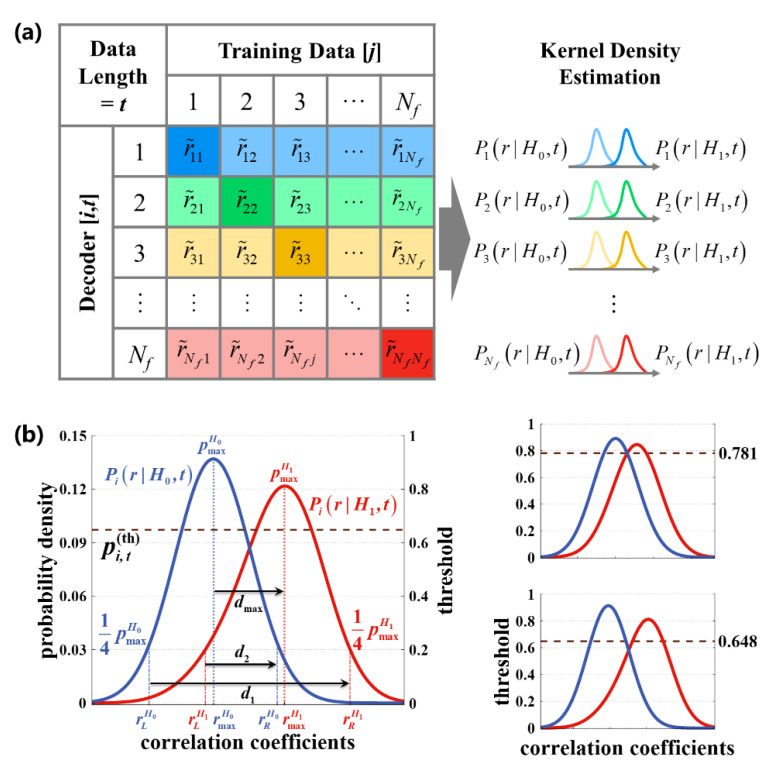
Key process of constructing a model for dynamic stopping: (**a**) the estimation of probability density functions for each target, and (**b**) the method to calculate the adaptive thresholds.

**Figure 5 sensors-20-04186-f005:**
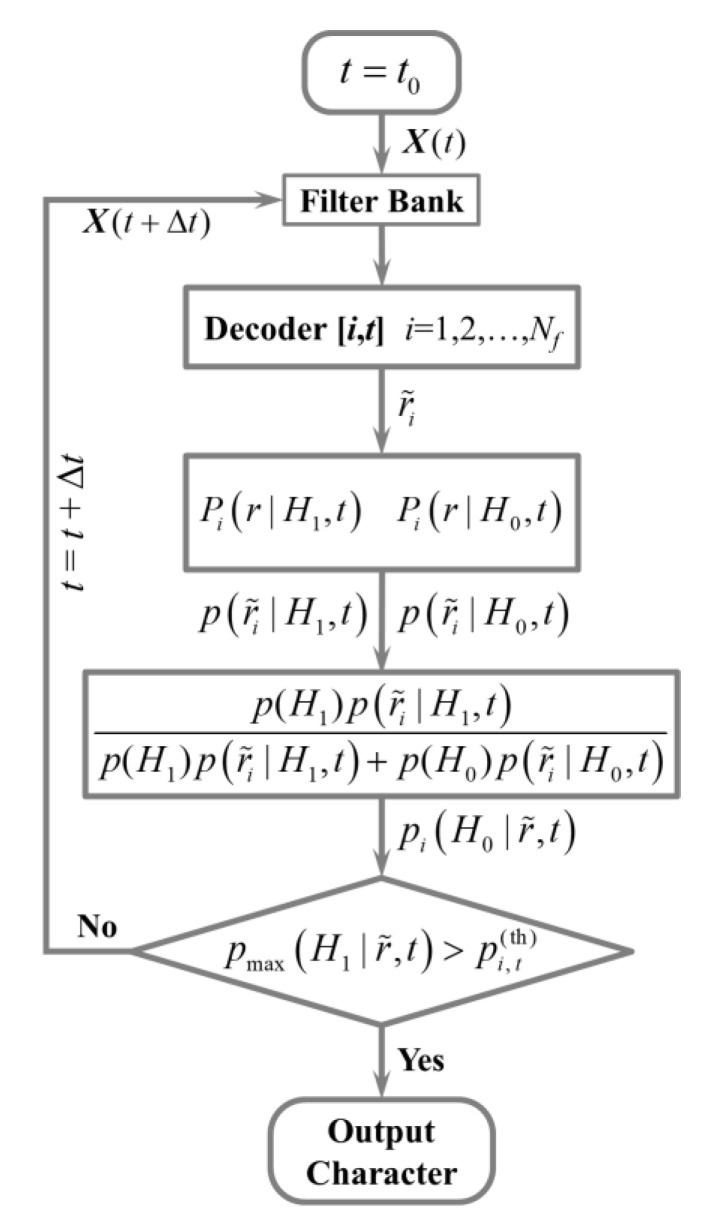
The flowchart of online recognition with dynamic stopping (DS) strategy.

**Figure 6 sensors-20-04186-f006:**
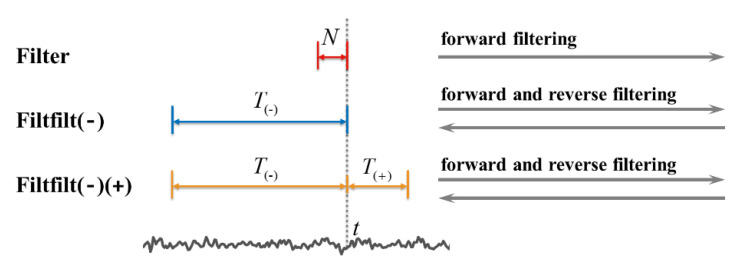
The schematic of the three filtering strategies tested in this study.

**Figure 7 sensors-20-04186-f007:**
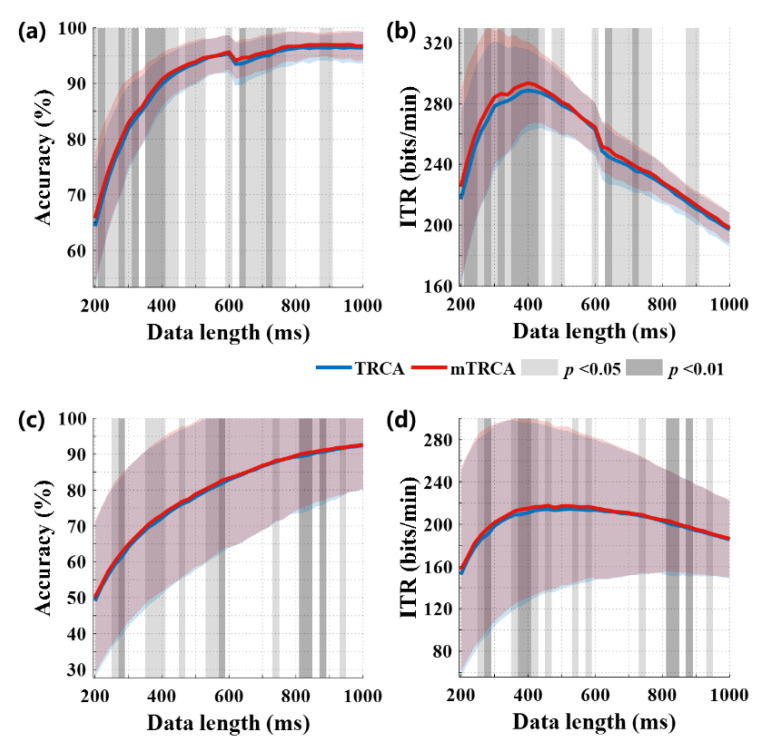
The recognition accuracies (**a**,**c**) and information transfer rates (ITRs) (**b**,**d**) using data from the offline experiments (**a**,**b**) and the benchmark dataset (**c**,**d**), respectively. The lines with deep color represent the averages while the shadings with light color represent standard deviation. The grey shading shows the significance of difference between accuracies of two spatial filters (Wilcoxon signed-rank test).

**Figure 8 sensors-20-04186-f008:**
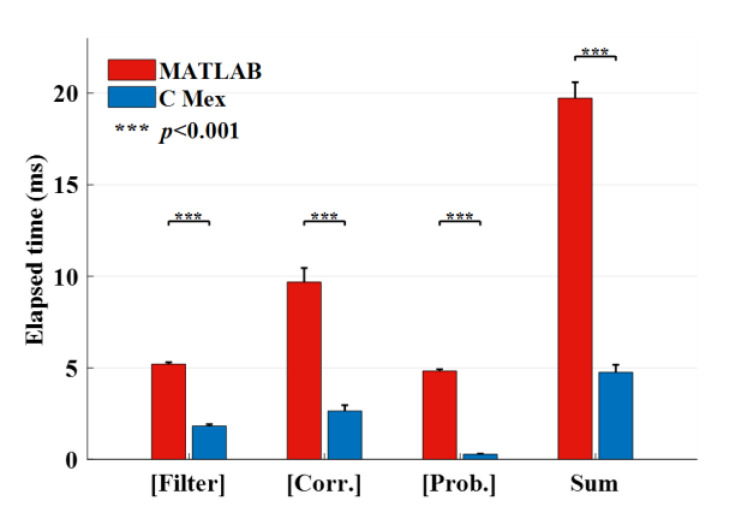
The average elapsed time of three online recognition procedures using MATLAB and C Mex. The right-most two bars are the sums of the three procedures. The asterisks indicate the significance (paired *t*-test). All the data for this comparison satisfy normal distribution (Lilliefors test, *p* > 0.05).

**Figure 9 sensors-20-04186-f009:**
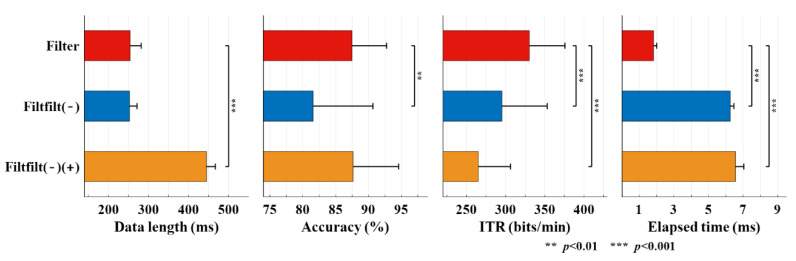
The data length, accuracies, ITRs and elapsed time under the three filtering strategies. The asterisks indicate the significance (Wilcoxon signed-rank test).

**Table 1 sensors-20-04186-t001:** Performance of cued-guided online experiments.

Subject	FS (Maximal Offline ITRs)			DS (Online ITRs)		
	Length (s)	Accuracy (%)	ITR (bits/min)	Length (s)	Accuracy (%)	ITR (bits/min)
S1	0.380	83.5	259.6	0.271	82.9	292.6
S2	0.440	94.8	303.3	0.255	84.0	305.3
S3	0.380	90.4	297.3	0.250	84.4	309.9
S4	0.440	87.5	262.8	0.291	83.0	285.7
S5	0.420	92.5	296.2	0.247	85.5	317.8
S6	0.280	90.3	334.5	0.223	93.0	380.6
S7	0.360	87.9	289.6	0.248	87.0	327.1
S8	0.260	92.9	361.5	0.225	98.0	420.2
S9	0.300	90.6	327.9	0.229	93.0	377.7
S10	0.440	86.5	257.5	0.316	80.0	260.6
S11	0.420	91.9	292.6	0.261	90.0	340.9
S12	0.280	88.9	325.5	0.236	89.0	345.9
Ave ± Std	0.367 ± 0.069	89.8 ± 3.1	300.7 ± 32.2	0.254 ± 0.028	87.5 ± 5.2	330.4 ± 45.4

**Table 2 sensors-20-04186-t002:** Performance of online free spelling experiments.

Subject	Trial Length (s)	No. of Trials	Spelling Rate	ITR
	(Gaze Shifting + Stimulating)	(Correct/Incorrect)	(cpm)	(bits/min)
S1	1.279 (1.0 + 0.279)	57 (48/9)	39.5	180.9
S2	1.262 (1.0 + 0.262)	89 (66/23)	35.3	148.9
S3	1.276 (1.0 + 0.276)	85 (64/21)	35.4	150.9
S4	1.283 (1.0 + 0.283)	95 (73/22)	35.9	155.2
S5	1.455 (1.2 + 0.255)	93 (77/16)	34.1	154.7
S6	1.047 (0.8 + 0.247)	73 (60/13)	47.1	212.3
S7	1.264 (1.0 + 0.264)	120 (92/28)	36.4	156.9
S8	1.287 (1.0 + 0.287)	54 (51/3)	44	220
S9	1.315 (1.0 + 0.315)	81 (66/15)	37.2	166.6
S10	1.308 (1.0 + 0.308)	97 (73/24)	34.5	147.1
S11	1.290 (1.0 + 0.290)	61 (52/9)	39.6	183.2
S12	1.248 (1.0 + 0.248)	61 (50/11)	39.4	177.3
Ave ± Std	-	-	38.2 ± 4.0	171.2 ± 24.5

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
