# Peer review of "Optimizing SSVEP-Based BCI System towards Practical High-Speed Spelling"

_sensors, 2020, doi:10.3390/s20154186_

Round 1
Reviewer 1 Report
This is well-written and very interesting paper. Of course, the study is connected to a very specialist area: the practicability of the brain-computer interface spellers based on steady-state visual evoked potentials in high-speed spelling. Yet, I am sure that such topic will be interesting for Sensors' readers. However, I expect Authors to broaden discussion by put some psychological aspects of using BCI, as well as, and more broadly, interacting with machines, to show that when we develop such (BCI-based) practical applications (here a system with portable EEG acquisition), we have to think also about developing some psychological programs for patients/students who will use BCI-tools and interact with machines. You need to consider "friendly interaction" not only in the technological sense. One paragraph with such psychological implications (or maybe some recommendations) of the results will be enough, but with good, fresh references, such as:
- Chaudhary, U. et al. (2020). Neuropsychological and neurophysiological aspects of brain‐computer‐interface (BCI) control in paralysis. The Journal of Physiology.
- Fritz, A. et al. (2020). Moral agency without responsibility? Analysis of three ethical models of human-computer interaction in times of artificial intelligence (AI). De Ethica, 6(1), 3-22.
- Klein, E. (2020). Ethics and the emergence of brain-computer interface medicine. In Handbook of Clinical Neurology (Vol. 168, pp. 329-339). Elsevier.
- Klichowski, M. (2020). People copy the actions of artificial intelligence. Frontiers in Psychology, 11:1130.
- Kogel, J. et al. (2020). What is it like to use a BCI?–insights from an interview study with brain-computer interface users. BMC Medical Ethics, 21(1), 1-14.
Forgetting the parallel development of technology and education/therapy has very negative consequences (see one of the studies listed above). That is why an interdisciplinary approach is necessary even in such technical studies as described in your paper.
Reviewer 2 Report
The paper proposes a SSVEP-BCI system for high-speed spelling that is optimized for practical use. The EEG amplifier, the mTRCA spatial filter, and the adaptive threshold generating method for the DS strategy are new contributions to the field. Also, the paper is very well written. I found the work very interesting and exciting. The following minor questions/recommendations should be addressed before publication.
- The capture of fig. 1 must briefly describe the interactions of the system’s elements from the beginning to the end of the process (the more detailed explanation given in the text is not sufficient).
- On line 71 it says: ‘The feasibility of real-time DS strategy in a practical online system needs further verification.’ Does the work presented in this manuscript contribute to such a verification?
- On lines 80-82 it says: ‘Firstly, the unfixed stimulating time in the DS situation results in different feelings for users compared with those in FS situation.’ What do the users feel with the FS strategy and with the DS strategy? Could learning by the users of the DS strategy help the optimization of the proposed BCI system?
- The last paragraph of the introduction should briefly highlight the novel contributions of this work.
- Did the corrected normal sight participants wear glasses or contact lenses? Could the use of different sight correctors influence the results?
- What information (besides the instructions of what to expect/how to interact with the BCI system) was given to the subjects before the experiment? For instance, were the subjects informed about the randomness of the stimulating times? Also, do the subjects drive cars regularly? Could the combination of awareness of what to expect before the experiment started, trained driving, and some learning during the 6 rounds of experiments influence the results?
- What is the ‘user datagram protocol (UDP)’ mentioned on line 103?
- On line 140 it says: ‘The lower right of figure 1 shows the circuit of the amplifier developed in this study.’ This appears to be a novel contribution and must be highligthed in the introduction and properly mentioned in the capture of fig. 1.
- On lines 175-176 it says: ‘… it is worthy minimizing the correlation of the current frequency and other frequencies.’ Please briefly explain why this minimization is worthwhile.
- On line 179, it says that the covariance matrix is ‘optimized as’. There is no optimization here. Formula (7) is just a definition for a modified covariance matrix. Calling it a ‘modified covariance matrix’ might be better.
- Please write the mathematical formula for the modified eigenvalue problem and remove the text from line 181 ‘could be derived from (4) as the eigenvectors’ (it is misleading). It should look like problem (4) but using the matrices S’ and Q’ instead.
- Please check the expression for the standard deviation given in formula (11). It appears that the power 2 is missing in the numerator.
- On line 210 it says: ‘This study proposed an adaptive threshold generating method by utilizing the pdfs of each decoder.’ Again, this appears to be a novel contribution and should be properly highlighted in the introduction.
- In fig. 5 showing the flowchart, please remove the plots from the third rectangle and the question mark from the rhombus (the rhombus stands for decision making and hence the question mark is redundant here).
- On lines 272-273 it says: ‘Wilcoxon signed rank test was used to compare the performance of the two methods.’ Please briefly explain the main advantage of using this specific test.
Round 2
Reviewer 1 Report
Dear Authors,
Thank you for all changes done in this context. I accept this version of MS with full conviction.
All the best,